# Preliminary Study on Application and Limitation of Microbially Induced Carbonate Precipitation to Improve Unpaved Road in Lateritic Region

**DOI:** 10.3390/ma15207219

**Published:** 2022-10-17

**Authors:** Sojeong Kim, Yeontae Kim, Suhyung Lee, Jinung Do

**Affiliations:** 1Department of Ocean Civil Engineering, Gyeongsang National University, Tongyeong 53064, Korea; 2Department of Highway and Transportation Research, Korea Institute of Civil Engineering and Building Technology, Goyang 10223, Korea

**Keywords:** unpaved road, lateritic soi, MICP, CaCO_3_, surface spraying, mixing, implementation

## Abstract

Some road systems are unpaved due to limited governmental finance and fewer maintenance techniques. Such unpaved roads become vulnerable during heavy rainy seasons following restrained accessibility among cities and traffic accidents. Considering the circumstances, innovative and cost–effective approaches are required for unpaved roads. Microbially induced carbonate precipitation (MICP) is an emerging soil improvement technology using microbes to hydrolyze urea generating carbonate ions, and precipitates calcium carbonate in the presence of calcium ion. Induced calcium carbonate bonds soil particles enhancing stiffness and strength when the MICP reaction takes place within the soil system. This study introduces the use of microbes on unpaved road systems consisting of in situ lateritic soils. The MICP technology was implemented to improve soil strength through two approaches: surface spraying and mixing methods. A series of soil testing was performed with varying chemical concentrations to measure precipitation efficiency, strength, and quality for construction material and see the feasibility of the proposed methods. The laboratory test results indicated that the surface spraying method provided improved; however, it was highly affected by the infiltration characteristics of used soils. The mixing method showed promising results even under submerged conditions, but still required improvement. Overall, the proposed idea seems possible to apply to improving unpaved road systems in the lateritic region but requires further research and optimization.

## 1. Introduction

An unpaved road means a road with an untreated surface consisting of soils and gravels less than 3 cm in particle size. In South Korea, the entire length of roads are 112,977,105 km, while paved roads are 98,683,177 km (87.35%), unpaved roads are 6,145,048 km (5.44%), and roads under construction are 8,148,880 km (7.21%) [1]. Metropolitan cities are covered with 100% paved road; however, the pavement ratio in country–level cities is 80.8%. The proportion of unpaved roads becomes higher in developing countries. For example, Laos has an unpaved road length of 39,214 km (76%) out of a total of 51,597 km [2]. The values imply the proportion of unpaved roads can be the indicator of the development rate.

Unpaved roads become liquefied when it rains. Due to the self–weight of vehicles, transportation systems are subjected to being stuck and idling. The braking distance on a wet surface is increased four times more than dry condition owing to reduced friction between tires and surface. When dry, fugitive dust is generated while driving [3]. Pebbles from the road are splashed. As most of the unpaved roads are uneven, driving efficiency is low. In addition, it is difficult to mark something on an unpaved road surface, requiring the installation of a separate sign. Comprehensively, the probability of car accidents increases [4].

Microbially induced carbonate precipitation (MICP) is a biochemical phenomenon that uses bacterial metabolism to hydrolyze urea forming carbonate ion and precipitate calcium carbonate (CaCO_3_) in the presence of calcium ions [5,6]. Precipitated CaCO_3_ bonds particles inducing the improvement of soil engineering properties (e.g., stiffness, strength) [7,8]. Unlike synthetic cement, which requires several days to cure, MICP takes approximately one day to cure [9]; therefore, it is time effective and eco-friendly. The MICP solution consists of water, bacteria, urea, and calcium. The solution shows a viscosity similar to water [10], allowing to easily infiltrate the solution into the ground without soil disturbance.

The MICP technology can be implemented via a surface spraying method, an injection method, and a mixing method. The surface spraying method involves spraying the MICP solution on the soil surface [11,12,13]. This method is the easiest way to implement MICP as heavy machinery is unnecessary and merely requires spraying the solution on the soil surface. However, if the solution does not infiltrate into the ground in an appropriate time, the cementation of the ground is not secured [14]. The injection method uses a hydraulic head to transport the MICP solution [15]. The MICP solution is injected via an injection well at the zone near the target ground and is dissipated as time passes or is artificially transported by an extraction well [16,17]. This method ensures a deeper depth to be treated; however, the wells should be installed near the target ground. Lastly, the mixing method replaces cement agent to the MICP solution in the case of the soil–cement mixing (SCM) method. When mixing geomaterials, such as field soils with sand and gravel, the MICP solution can be mixed instead of using synthetic cement [18,19,20]. Unlike other methods, of which there are a multiple treatments available, the mixing method is a one-shot treatment. Fine soils or well-graded soils are known to be unfavorable on the MICP application due to low porosity and hydraulic conductivity [21]. In this paper, the surface spraying method and the mixing method are two feasible approaches using field soils (well-graded lateritic soil) in the aim of improving unpaved roads.

This paper examines the feasibility of applying MICP to improve unpaved roads using field soils. The treatment approaches are a surface spraying method and mixing method. A batch of tests was conducted to measure the efficiency of CaCO_3_ precipitation varying with bacterial and chemical concentrations. A series of soil column tests were performed to check the strength improvement of treated specimens. A modified CBR test was carried out for treated specimens to check the applicability as road material. A microscopic examination was carried out to analyze the cementation pattern of treated samples. At the end of the paper, the limitations based on the results are discussed and comprehensive comments are provided.

## 2. Materials and Methods

### 2.1. Test Soils

Three specific soils were used in this study. A field soil, standard sand, and white pebble. For field soils, soils were collected from the construction site for an industrial complex at Tongyeong, South Korea. The collected soils were oven–dried and several basic tests were performed. Figure 1 shows the particle size distribution enclosed in the photos of used soils and basic engineering properties are summarized in Table 1 (*D*_50_: mean particle diameter, fine: passing through No. 200 sieve, *G*_s_: specific gravity, *w*_L_: liquid limit, *w*_P_: plastic limit, *e*_min_: maximum void ratio, and *e*_max_: minimum void ratio). The field soils are classified as SW–SM (sandy soil with good particle size distribution including silt) based on the unified soil classification system (USCS) [22,23,24,25]. The maximum dry unit weight *γ*_d(max)_ = 16.19 kN/m^3^ under modified compaction energy and the optimum water content *w*_opt_ = 21%. The field soils seem lateritic based on color, properties, and geologic locations.

Jumunjin standard sand was used as a fine aggregate. The sand is classified as SP (poorly graded sandy soil) based on USCS. The particle size distribution and a photograph are shown in Figure 1 and engineering properties are shown in Table 1.

Commercial white pebble for aquarium purpose was used as a coarse aggregate (Blue Marine Partner), which has 3–5 mm in diameter considering a boundary and scale effect with the testing system. The pebble is SP, which is the same as the standard sand; however, the *D*_50_ of the pebble is 5.7 times larger than one particle of sand (Figure 1 and Table 1). Pebbles were washed with water to remove the dust on the pebble surface and dried before use.

This paper implemented the MICP technology using the surface spraying method and mixing method. The next section describes how specimens were selected for these methods.

#### 2.1.1. Sample for Surface Spraying Method

The surface spraying method refers to a ground improvement method spraying the MICP solutions on the soil surface. The MICP solution penetrates from the surface into the ground by gravitational drainage. The solution in the ground is located between the particles by capillary force where CaCO_3_ forms. Cheng et al. [26] reported that the engineering performance of MICP–treated samples under unsaturated conditions was shown to be better and more efficient than that under saturated conditions. The surface spraying method is expected to be effective in improving unpaved roads because the unpaved surface usually has no obstacles and most unpaved roads are unsaturated.

When this surface spraying method is implemented, the most important factor is the infiltration rate (*f*) of the ground [27]. The *f* refers to the one-dimensional dissipation rate of the hydraulic head by gravity. Assuming the field soils will be moderately improved by mixing fine aggregates, the *f* of field soil-standard sand mixtures were measured with respect to the mass ratios of 100:0, 75:25, 50:50, 25:75, and 0:100. After mixing the dried soils with the mass ratio, the mixtures were placed in a soil column measuring 5 cm in diameter and 10 cm in height. Standard compaction energy of 5.9 kg∙cm/cm^3^ was applied when preparing the specimens (e.g., 196.35 cm^3^ volume, 2–layered compaction, 25–time blows per layer, hammer of 1.158 kg, and 20 cm drop height) [28]. The mold had a collar of 3 cm in height at the top of the mold. Once the specimen preparation was completed, water was poured into the zone of the collar and the drawdown of water was measured with time.

The experiment results are shown in Figure 2a, and the infiltration rate according to the mixing ratio is summarized in Figure 2b. The slope of the infiltration test result is defined as *f* (cm/s). The *f* of mixtures showed a substantial variation according to the mixing ratio. The pure field soil had the *f* of 0.0015 cm/s, then the *f* lowered to 0.0006 cm/s when the standard sand was mixed 25%. It can be said that the compaction efficiency would be enhanced by adding 25% sand, and subsequently, the *f* decreased. Thereafter, as the standard sand increased to 50%, 75%, and 100%, *f* increased.

The surface spraying method requires a sufficient level of *f*, but the *f* was too low to test as 0.0015, 0.0006, and 0.0014 cm/s, before the mixing ratio of standard sands reached more than 75% (Figure 2a). At least 75% of the portion of sand was determined as a minimum ratio to perform a test in this study. Therefore, in this study the mixture of 25% field soil + 75% standard sand was used in the case of the surface spraying method. The mixture of 25% field soil + 75% standard sand showed a maximum dry unit weight (*γ*_d(max)_) of 17.6 kN/m^3^ and an optimum water content (*w*_opt_) of 15.6% under standard compaction energy. It is classified as SP by USCS. This mixture is herein called a 1:3:0 sample (mass ratio of field soil: fine aggregate: coarse aggregate). The particle size distribution is shown in Figure 3.

#### 2.1.2. Sample for Mixing Method

In general cases, soil improvement is often achieved through a soil–cement mixing (SCM) method. The field soil, fine aggregate, and coarse aggregate are mixed with cement. In the same manner, this study examined whether the ground is improved if the MICP solution is used instead of pure water with no cement when mixing soils. The combinations of field soil–fine aggregate–coarse aggregate mixture are infinite cases, so a simplification was applied. The mass ratio of 1:1:1 (field soil: fine aggregate: coarse aggregate) was used in the case of the mixing method. The identical mass of the three soils was mixed. The mixture is classified as SP; however, the mixture locates at the boundary of SW (well–graded sandy soil) by USCS. *W*_opt_ = 13% and *γ*_d(max)_ = 19.3 kN/m^3^ under standard compaction energy, while *W*_opt_ = 11% and *γ*_d(max)_ = 19.8 kN/m^3^ under modified compaction energy. This mixture is herein called a 1:1:1 sample, and the particle size distribution is shown in Figure 3.

### 2.2. Implementation

#### 2.2.1. Specimen Preparation for Soil Column

A soil column measuring 5 cm in diameter and 10 cm in height was manufactured to prepare specimens for strength evaluation. The soil column is made of 5 mm-thick acrylics and can be vertically separated on both sides so that the specimens within the column can be taken out without any damage after treatment. Vacuum grease was applied between the split column to prevent leakage. The column was tightened through a hose clamp along the column surface. At the bottom of the soil column, a bottom plate with drainage holes was placed holding the split column. At the top of the soil column, a collar of 3 cm in height was placed to allow space for treatment solutions. An example of the soil column can be seen in Figure 2.

The 1:3:0 and 1:1:1 samples were oven-dried before use. Samples were placed in the split column using standard compaction energy in the same manner as the infiltration testing. Assuming the surface spraying method will be implemented for less compacted ground, the specimens which will be treated by the surface spraying method were compacted with a water content (*w*) of 10%, which corresponds to the relative compaction of 90% on the dry side. Assuming the mixing method will be implemented for the height density, the specimens, which will be treated by the mixing method were compacted with *w*_opt_ = 13%. A detailed description of the specimen preparation will be followed. All specimens were duplicated.

#### 2.2.2. MICP Solutions

The MICP solutions in this study are divided into a microbial solution and a cementation solution. First, the microbial solution refers to cultured bacteria in a growth media. A growth media containing Tris buffer of 15.79 g/L, yeast extract of 20 g/L, and (NH_4_)_2_SO_4_ of 10 g/L with the solvent of deionized water was prepared. The bacteria used was *Sporosarcina pasteurii* (Korean Collection for Type Cultures, KCTC 3558). Frozen bacteria of 4 mL were added in the growth media and cultured under 200 rpm at 30 °C in a shaking incubator. The cultured media was cultured in a growth media until the optical density of the cultured media with a wavelength of 600 nm (*OD*_600_) reached ~1.5. The *OD*_600_ was measured using a spectrophotometer.

The cementation solution refers to a solution in which equimolar urea and calcium chloride (CaCl_2_) were mixed in deionized water. The concentrations used in the study were 0 M (untreated), 0.25 M, 0.5 M, 0.75 M, 1.0 M, and 2.0 M. Different concentrations were used depending on the purpose of the experiment. Immediately before the specimen treatment, the microbial solution and the cementation solution were mixed in a volume ratio of 1:5 and then used for the treatment. The mixed solution is called the MICP solution.

#### 2.2.3. MICP Treatment

The MICP treatment strategy differed depending on the surface spraying method and the mixing method. First, in the case of the surface spraying method, it was intended to treat the entire void volume of the 1:3:0 specimen. The total void volume is called 1 pore volume (1 PV). The 1 PV as 80–90 cm^3^ was calculated for specimens. Therefore, the MICP solution of 90 mL was sprayed on the specimen at a spraying rate of 0.01 cm/s. The value of the spraying rate is equivalent to spraying 12 mL per minute on a 5–cm–diameter soil column. The treated specimens were compared with the untreated ones. One or three MICP treatments were implemented per day. The cementation solutions of 1 M and 2 M were used for the MICP solution.

For the mixing method, the MICP solutions were mixed with a 1:1:1 sample with *w*_opt_ = 13%. The *w*_opt_ = 13% is equivalent to 96% degree of saturation. Thus, the *w*_opt_ is intended to fill most of the void in the specimen. Assuming the mixing condition can use more diverse concentrations than the surface spraying method, the concentrations of 0, 0.25, 0.5, and 1.0 M cementation solutions were used. Since the mixing takes place one time, all cases correspond to a single MICP treatment. The treated specimens were cured for two days at room temperature before the strength test. The scheme of the experimental procedure is briefly shown in Figure 4.

### 2.3. Evaluation

#### 2.3.1. Precipitation Efficiency

Bacteria hydrolyze urea generating carbonate ions, which react with calcium ions and form calcium carbonate precipitation. The role of bacteria not only hydrolyzes urea. The surface of bacteria is negatively charged, so calcium cation gathers around bacteria. Thus, bacteria seem to be carrying calcium ions. In this situation, bacteria hydrolyze urea forming carbonate ions. The formed carbonate ion immediately reacts with the calcium ions, which are attached to bacteria following the bacterial encapsulation within CaCO_3_ crystals occurring during precipitation. Therefore, as the reaction proceeds, the bacterial population decreases [29,30]. The encapsulation depends on the concentration of the used solute.

By comparing the amount of CaCO_3_ precipitation by experiment to one by theory with respect to concentrations, the precipitation efficiency of CaCO_3_ can be evaluated according to the concentrations. A series of beaker tests were performed to validate the precipitation efficiency. The bacterial solution of 20 mL and the cementation solution of 100 mL were prepared. The bacterial solution had *OD*_600_ ~1.0 and ~2.0. The cementation solution had a concentration range of 0.1 M to 1.0 M. The bacterial solution and the cementation solution were mixed and left at room temperature for 3 days to allow for sufficient time to react. After 3 days, the supernatant of the mixture was removed, the beaker was dried, and the precipitation at the bottom of the beaker was measured. The theoretical molar mass of CaCO_3_ is 100.0869 g/mol. The precipitation efficiency was calculated by the actual amount over the theoretical amount in percent. For example, if CaCO_3_ of 10 g is experimentally precipitated when CaCO_3_ of 20 g is theoretically expected, the precipitation efficiency is calculated as 50%.

#### 2.3.2. Strength

The strength of the treated specimens was evaluated by a uniaxial compression test. The treated specimens moved from the split column to the pedestal of the uniaxial compression testing machine. A strain rate of 1 mm/min, which corresponds to 1% axial strain of specimen, was applied during axial compression. After shearing up to ~20% axial strain, samples of ~20 g were collected from the sheared specimen to measure the amount of CaCO_3_ precipitation.

In the case of the specimens treated by the surface spraying method (e.g., 1:3:0 sample), a sufficiently perceptible level of cementation occurred on the soil surface; however, the level of cementation lowered as the depth increased due to insufficient infiltration rate, and clogging occurred at the sprayed surface. Accordingly, when the uniaxial compression test was conducted, the less cemented lower height of the specimen was subjected to shearing rather than the more cemented upper height of the specimen. To overcome the underestimation and evaluate properly, a surface resistance of the treated specimen was necessary instead of whole shear resistance (i.e., uniaxial compressive strength). The surface resistance was evaluated by a pocket penetrometer (E–280, Addag, Aachen, Germany). A grooved prove with 5 mm in diameter is attached at the tip of the penetrometer. The groove is at 6.4 mm from the prove tip. A calibrated spring is connected to the prove within the body of the penetrometer. For use, the pocket penetrometer is placed at the center of the surface and perpendicular to the surface, and the penetrometer is pushed into the ground until the penetration depth is achieved as much as 6.4 mm. The penetration resistance is manually marked along the penetrometer body. The range of the penetration resistance is 0–441.3 kPa. Assuming the penetration resistance to be an undrained shear strength of the specimen, the uniaxial compressive strength of the specimen can be calculated by doubling the measured penetration resistance. Considering the column diameter of 50 mm, the boundary effect between the column wall and the penetration diameter can be negligible [31].

In the case of the treated specimen by the mixing method (e.g., 1:1:1 sample), a modified CBR test was performed on untreated/MICP–treated specimens for clearer evaluation as road materials [32]. When preparing the specimens, 10, 25, and 55 blows per layer were used with 5 layers as the maximum particle size of the sample was ~5 mm. An absorption expansion test was performed prior to a penetration test.

#### 2.3.3. Microscopic Analysis

A chunk of treated specimens after failure was obtained for scanning electron microscopy (SEM) to observe the microscopic image of the samples. The collected samples were gold-coated before the examination to enhance the resolution of the image. The untreated samples, treated samples by the surface spraying method, and sample by the mixing method were examined.

#### 2.3.4. Quantification of Cementation Level

After the strength testing, a chuck of the failed sample was collected to quantify the level of cementation. The collected sample was oven-dried and dissolved in 1 M HCl. CaCO_3_ reacts with HCl and becomes CO_2_ showing bubbles. Once the reaction (bubbling) no longer occurred, the supernatant was removed and oven-dried. The mass of CaCO_3_, *m*_c_ is defined as the ratio of CaCO_3_ over the pure sample in percent. The *m*_c_ = 1% means 1 g of CaCO_3_ is formed in 100 g of soil.

## 3. Results

### 3.1. Precipitation Efficiency with Respect to Recipes

Figure 5 shows the CaCO_3_ precipitation efficiency with the microbial concentrations of *OD*_600_ = ~1.0 or ~2.0 and the cementation concentrations of 0.1 M, 0.3 M, 0.5 M, 0.7 M, and 1.0 M. When the bacterial density was relatively low (*OD*_600_ = ~1.0), ~80% CaCO_3_ was formed at 0.1 M, but the precipitation efficiency decreased as the concentration increased. Seemingly, 80% CaCO_3_ was the maximum precipitation at the given chemical. When the solute concentration was 1.0 M, only about 40% of CaCO_3_ was precipitated compared to the maximum possible precipitation. It can be said that as the solute concentration increases, the bacterial encapsulation is facilitated following the decrease in the number of bacteria, and urea hydrolysis is not fully performed even though there is excess urea.

For *OD*_600_ = ~2.0, which is a relatively high bacterial density, the precipitation efficiency generally shows ~80% regardless of the concentrations. This observation indicates that the bacterial density of *OD*_600_ = ~2.0 would be sufficient to hydrolyze urea than one of *OD*_600_ = ~1.0 even if there is a bacterial encapsulation phenomenon within the given range. Note that low efficiency does not mean low total precipitation. For example, with *OD*_600_ = ~1.0, the solute concentration of 1.0 M with 40% efficiency (i.e., 0.4 M CaCO_3_) will show higher CaCO_3_ than one of 0.5 M with 60% efficiency (i.e., 0.3 M CaCO_3_).

In addition, the rate of CaCO_3_ precipitation is different depending on the concentration. The rate of precipitation is related to the mineralogy of CaCO_3_ precipitates [33] and the distribution of CaCO_3_ along the direction of injection. Therefore, it is very important to see the MICP reaction with the samples in practice. The following section explains the results of soil column testing.

### 3.2. Strength of MICP-Treated Specimen

#### 3.2.1. Specimen Treated by Surface Spraying Method

There are a couple of implementing MICP by the surface spraying method: one-phase and two-phase spraying. The two-phase method is that bacterial solution is injected first (or uses indigenous bacteria), allowing sufficient retention time to attach bacteria onto the particle surface, and then cementation solution is sprayed. CaCO_3_ precipitation takes place inside the soil matrix. The two-phase spraying may cause the attached bacteria to be washed down together while infiltration takes place. In this situation, the precipitated CaCO_3_ accumulates at a certain depth, but shows less cementation on the soil surface. In addition, the implementation time would increase as the spraying should be conducted in two phases. On the other hand, the one-phase method sprays the mixture of bacteria and cementation solution immediately after mixing. Although the spraying would be less than the two–phase method, micro–crystals of CaCO_3_ start to form once mixing. This phenomenon sounds similar to injecting micro CaCO_3_ precipitates during the spraying of MICP solution. Such fine CaCO_3_ causes clogging at the zone of the injection port, resulting in the decrease in *f* and the increase in spraying time. This process facilitates clogging. Therefore, sufficient *f* is very important for the target ground to implement MICP in the case of the one-phase method.

The 1:3:0 specimen for the surface spraying method showed the *f* of 0.01 cm/s. After completing the surface spraying method, the soil surface of the specimen seemed heavily cemented based on visual observation and slight touch. However, when the specimen was removed from the split column to perform the uniaxial compression test, the uniaxial compressive strength (UCS) of the treated specimen showed to be similar to one of the untreated specimens. During the shearing phase, the lower part of the specimen failed and no damage was found at the upper part of the specimen. The observation implied the cementation was localized at the zone near the sprayed surface and less cementation was made at the lower part of the specimen. Uniform cementation was not achieved along the specimen height due to the insufficient infiltration rate resulting in localized cementation on the surface. Therefore, an alternative measure, a pocket penetrometer test, was attempted to evaluate the improvement of the cemented surface.

Figure 6 shows the converted UCS of the 1:3:0 specimens treated by the surface spray method. The two-time multiplied reading of the pocket penetrometer was defined as the converted UCS. The converted UCS of the untreated specimen showed 108 kPa. The converted UCS of treated specimens were 687 kPa by 1 M and 353 kPa by 2 M MICP solution one time (Figure 6a). During the pocket penetrometer testing, the penetration resistance gradually increased at a low strain, but a rapid strain occurred following a punching shear failure when applied penetrating force was reached to the maximum resistance of specimen. This is because of the heavily cemented surface resists the penetrating force at the initial stage; however, once the penetrating force exceeds the maximum resistance of soil, a less cemented zone appears, and sudden punching shear failure takes place. It is interesting to note that the UCS by 2 M recipe was lower than the one by the 1 M recipe. Considering the precipitation efficiency according to concentrations (Figure 5), the results imply that the MICP recipe with high concentration does not always bring a high strength of soils. The specimens showed *m*_c_ ~0.5% regardless of the cases. Precipitation efficiency, clogging, infiltration rate, etc., are associated with the engineering performance of target soil [8]. There seems to be a complex relationship among solutions and soils.

For the triple treatments (MICP #3 in Figure 6a), the specimen treated by 1 M MICP solution showed the pocket penetration resistance of 441 kPa (upper limit, converted UCS = 882 kPa) was measured during ~1 mm penetration. The one treated by the 2 M MICP solution showed the converted UCS = 834 kPa. The specimen treated by the 2 M MICP solution showed a lower strength than the one by 1 M either #1 or #3 treatments.

The *f* of the field soil is 0.0015 cm/s. The f was artificially increased to 0.01 cm/s by mixing the standard sand of 300 wt.% Nevertheless, the results of the surface spraying method showed limited cementation, even for 10 cm height specimen. However, meaningful cementation is observed in the zone near spraying. Therefore, it can be confirmed that the *f* is a critical factor for implementing MICP with the surface spraying method. The *f* should be sufficient or the MICP recipe should be modified to achieve the homogeneity of cementation in the use of the MICP surface spraying method.

#### 3.2.2. Specimen Treated by Mixing Method

The specimens treated by the mixing method had no issue with the uniformity of cementation since the infiltration was not associated; therefore, all specimens were evaluated by a uniaxial compression test without using a pocket penetrometer (Figure 7a). The specimen treated by water (untreated) showed the peak stress of 340 kPa at a strain of ~1.5%. Specimens treated by 0.25 M and 0.5 M MICP solutions presented the peak stresses of 403 kPa and 391 kPa, respectively. Compared to the untreated specimen, the treated specimens show the UCS to be 19% and 15% higher. In fact, it is difficult to mention a dramatic improvement in strength that occurred due to treatment. Even for the specimen treated by the 1 M MICP solution, the UCS is lower than the untreated one. When preparing the specimens by the mixing method, a reaction time of 2 days was allowed and then oven-dried at 60 °C for 24 h to remove any chemical reaction; the same was the case for the untreated specimens. As shown in Figure 7b, the 1:1:1 sample was found to have a naturally high UCS without treatment. Therefore, it was meaningful to check the strength under wet conditions.

A modified CBR test was performed to evaluate the strength improvement of treated specimens including wet conditions (Figure 8). A 0.5 M MICP solution was used for the treatment. The modified CBR test includes an absorption–expansion test and a penetration test [32]. First, the expansion ratios of the untreated 1:1:1 specimens compacted by 10, 25, and 55 times per layer were 0.02%, 0.03%, and 0.1%, respectively (Figure 8). When the expansion ratio is less than 1%, it is regarded as a good roadbed [34], so the expansion characteristic of the untreated sample was shown to be naturally low. Nonetheless, the treated specimens showed a 0% expansion ratio during a 96-h submersion. No expansion infers the treatment induced the cementation of the sample.

Based on the Expressway Construction Guide Specification [34], a subgrade for embankment should be higher than the modified CBR of 10% at 95% relative compaction (RC 95%). The untreated specimens provided the modified CBR = 7.2% (Figure 8), which fails the construction specification of the subgrade. On the other hand, the 0.5 M MICP solution-treated specimens showed the modified CBR = 13.3% satisfying the specification. The post-CBR treated specimens showed *m*_c_ = ~0.2%. Even the specimen with *m*_c_ = ~0.2% demonstrated a perceptible strength improvement.

Overall, the untreated 1:1:1 sample has a naturally high UCS when dried. Thus, a wet test such as the modified CBR is necessary for the sample to properly evaluate the engineering performance. Specimens treated by 0.5 M MICP solution showed no expansion and an increase in the modified CBR compared to the untreated specimens. The performance was achieved even with the amount of CaCO_3_ = 0.2%. Therefore, if higher CaCO_3_ is precipitated in the specimen, higher performance is expected.

### 3.3. Microscopic Analysis

The 1:1:1 untreated sample (Figure 9a,b), 1:3:0 sample tri–treated by 1 M MICP solution and surface spraying method (Figure 9c,d), and 1:1:1 sample treated by 0.5 M MICP solution and mixing method (Figure 9e,f) were analyzed through SEM images. The 1:3:0 sample is the same as the one shown in Figure 6b. The white pebbles were not identified in the SEM images due to magnification (Figure 9a,b,e,f), whereas sand particles and clay clusters can be shown in the images. The samples treated by the surface spraying method (Figure 9e,f) had *m*_c_ = 0.2%, so a definite observation of CaCO_3_ was not specified in the images. However, the sample treated by the surface spraying method showed a bunch of CaCO_3_ precipitations with spherical shape. The spherical shape of CaCO_3_ is identified as vaterite [33,35]. Vaterite is formed when CaCO_3_ is precipitated at a fast rate. Therefore, the CaCO_3_ during the surface spraying method was formed fast, which resulted in a high level of clogging.

Based on the microscopic analysis, it is confirmed that the surface spraying method accompanies with the fast rate of precipitation forming vaterite. Meanwhile, the mixing method comes with an insufficient level of cementation, which is not identified via SEM image. In conclusion, recipe optimization seems critical to again implement the MICP approach over the soils in this study.

## 4. Discussion

### 4.1. Surface Spraying Method

The surface spraying method is an innovative method that does not require the disturbing ground to improve because the method only needs to spray a cementing solution on the surface. If the infiltration rate is allowed, the solution can be sprayed several times, which can create diverse ways of treatment. Once the bacterial solution and the cementation solution are mixed, CaCO_3_ starts to form immediately. Some of the immature CaCO_3_ precipitates are infiltrated into the ground and some are accumulated on the surface. Therefore, soils with a low infiltration rate would encounter localized cementation and clogging on the surface. In this study, the sample was made of 25% field soil and 75% standard sand; however, uniform or acceptable cementation failed even along the 10 cm–height specimen. The *f* of 0.01 cm/s was insufficient to implement the given MICP recipe. Note that different rate should be examined further.

Moreover, the surface spraying method using MICP can be effective if the localized surface is on purpose. Exemplarily, the method can be used for fugitive dust control. In South Korea, 44% of fine dust comes from unpaved roads, construction sites, and yards [36,37]. To control the fugitive dust, spraying water, dust barriers, covering geosynthetics, etc. are used. The surface spraying method using MICP would be a unique alternative to handle the fugitive dust issues. In this application, the clogging phenomenon during spraying would be useful rather than problematic.

### 4.2. Mixing Method

The biggest disadvantage of the surface spraying method is the difficulty in securing uniformity of cementation. The mixing method ensures the uniformity of cementation such as using synthetic cement. However, the biggest issue with the mixing method is that everything needs to be solved within one mixing. The modified CBR test showed a promising result of using MICP to improve unpaved roads. The amount of precipitated CaCO_3_ was merely ~0.2%. If the added 0.5 M MICP solution reacted 100%, *m*_c_ = 0.54% was expected. If 1 M MICP solution is used, *m*_c_ = 1.08% is expected. However, as confirmed in the efficiency test, the higher the concentration, the lower the efficiency.

There are several approaches to increase the level of cementation on the MICP technology: increases in bacterial density, the concentration of solutions, precipitation efficiency, amount of solution, and so on. A bacterial density of *OD*_600_ = 1.5 was used in this study. Based on a laboratory test (data not shown in detail), a bacterial growth curve showed a death phase when *OD*_600_ = 1.8. Thus, OD600 = 1.5 could be sufficiently fresh and regarded as high density. The cementation solution of 0.5 M was used in the mixing method. The increase in the cementation solution will increase in *m*_c_, but will affect the precipitation efficiency. Recently, there have been attempts to increase the precipitation efficiency by using an additive in the MICP solution [38,39,40,41]. A biopolymer, such as Xanthan gum, PVA, etc., can enhance a bacterial habitat favorable and the viscosity of solutions is increased resulting in higher retention against gravitational infiltration. Eventually, the use of additives can increase the amount of CaCO_3_ precipitation. In addition, optimum water content was used. This value exhibited a limited amount of CaCO_3_ to form. If *w* increases, the sample moves to the wet side of the compaction curve and the dry unit weight decreases. In this case, the natural strength of the specimen will be lowered, but the strength of the MICP–treated specimen would increase as *m*_c_ increases. Therefore, it can be said that optimizing the recipe is still an important task and needs further study in the mixing method.

## 5. Conclusions

In this study, for the improvement of lateritic unpaved roads using the MICP technology, two approaches were used: the surface spraying method and the mixing method. A series of batch tests were performed according to the recipes to understand the precipitation efficiency of CaCO_3_ by MICP. The strength improvement was confirmed by soil column tests and modified CBR tests. The cementation pattern was evaluated by microscopic analysis. The general conclusions are summarized as follows:The CaCO_3_ precipitated by the MICP process depends on the bacterial density and the solution concentrations. The higher the bacterial density and the lower the solution concentration, the higher the precipitation efficiency.The surface spraying method is easy to implement for MICP; however, the method requires a sufficient infiltration rate of the ground. When the infiltration rate is low, the uniformity of cementation is unfavorable due to clogging issues on the surface. If a specified surface treatment is needed, such as surficial dust control, the surface spraying method can be an attractive option to consider.The mixing method is a one-shot solution to cement. The MICP recipe has a great influence on the final performance of the target soil. The key is to properly optimize the recipe, such as the solution concentration and the amount of solution. It seems possible to enhance the cementation efficiency by using an additive.

The use of MICP is possible to improve the engineering performance of unpaved roads from a new perspective. This study focused on the MICP application using lateritic soils. However, this preliminary study showed that the most important task is to optimize the recipe in the consideration of ground characteristics, especially in the lateritic area. From a positive point of view, MICP could be implemented with the aim of improving unpaved roads. We hope the results of this study will help optimize recipes in future research.

## Figures and Tables

**Figure 1 materials-15-07219-f001:**
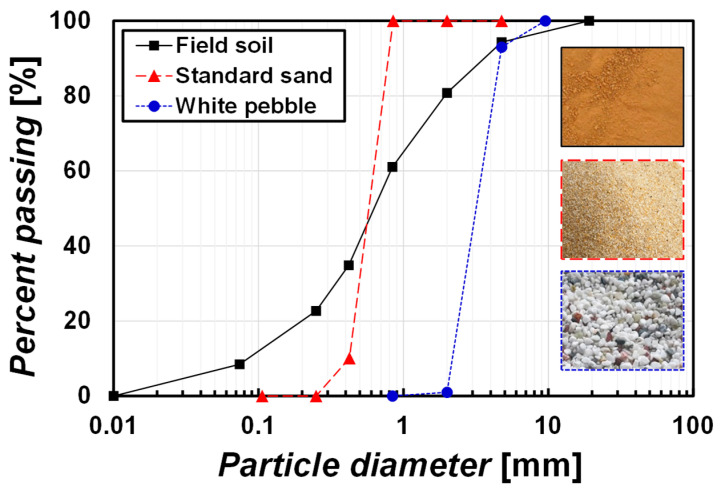
Particle size distribution of used soils and their photographs.

**Figure 2 materials-15-07219-f002:**
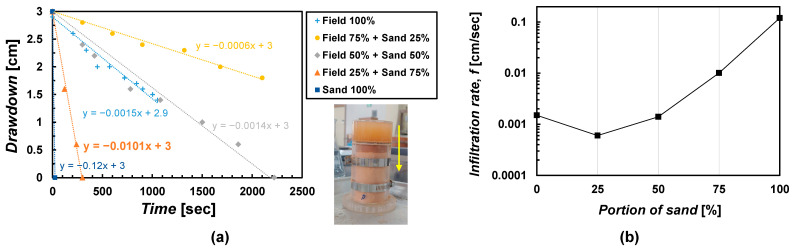
(**a**) Infiltration test results and (**b**) infiltration rates according to mixing ratio.

**Figure 3 materials-15-07219-f003:**
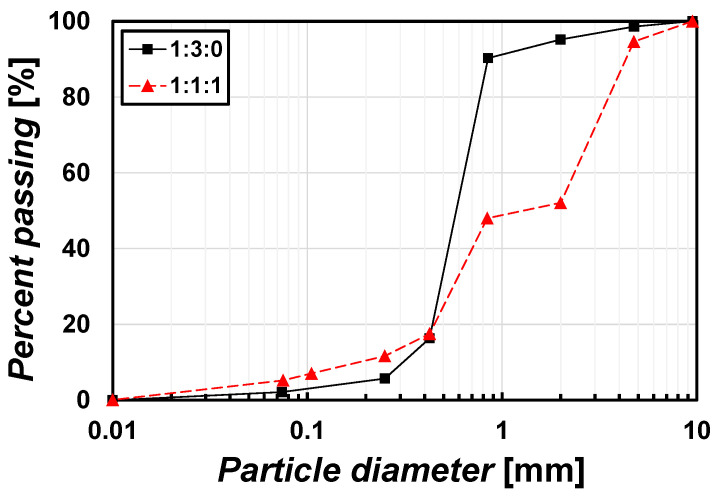
Particle size distribution of test samples.

**Figure 4 materials-15-07219-f004:**
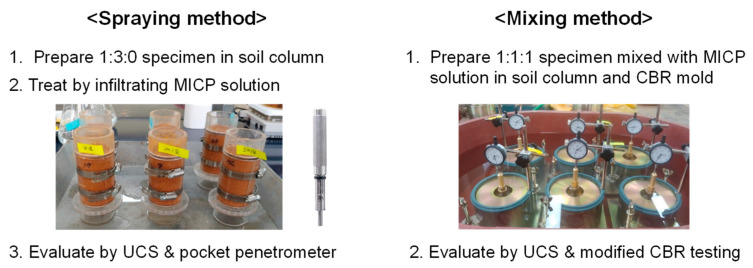
Brief scheme of experimental flow.

**Figure 5 materials-15-07219-f005:**
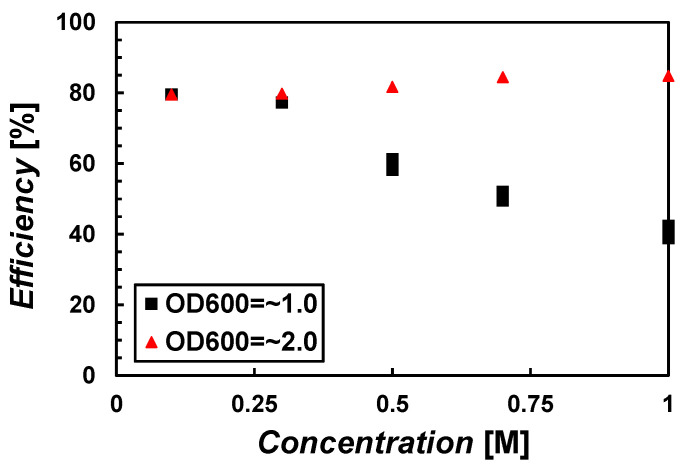
Precipitation efficiency with respect to recipes.

**Figure 6 materials-15-07219-f006:**
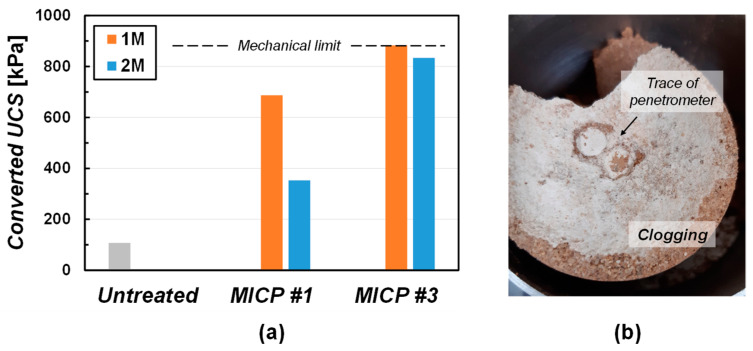
(**a**) Converted UCS from pocket penetrometer measurement and (**b**) cemented surface of the specimen treated by 1 M MICP solution three times.

**Figure 7 materials-15-07219-f007:**
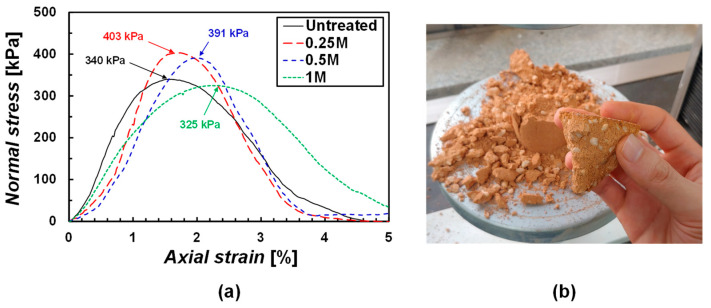
(**a**) Uniaxial compression test results for mixing method and (**b**) untreated specimen after compression test.

**Figure 8 materials-15-07219-f008:**
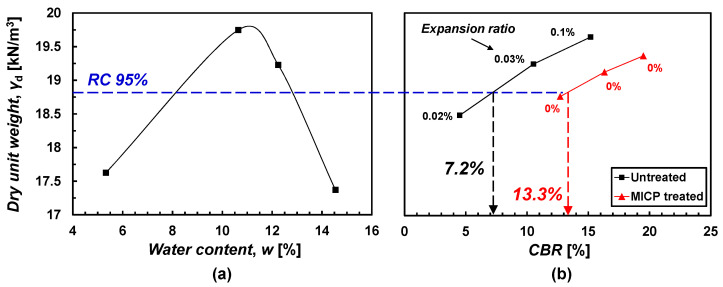
Modified CBR test results on untreated and MICP-treated specimens. (**a**) compaction curve and (**b**) CBR curves.

**Figure 9 materials-15-07219-f009:**
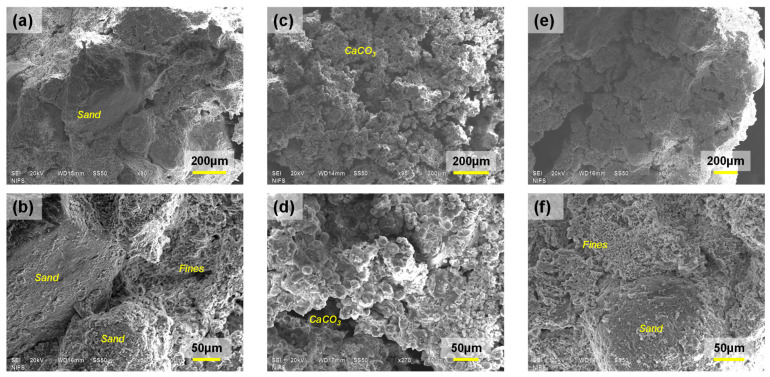
SEM images: (**a**,**b**) 1:1:1 untreated sample, (**c**,**d**) 1:3:0 sample tri–treated by 1 M MICP solution and surface spraying method, and (**e**,**f**) 1:1:1 sample treated by 0.5 M MICP solution and mixing method.

**Table 1 materials-15-07219-t001:** Engineering properties of used soils.

Soil Type	*D*_50_ [mm]	Fine [%]	*G* _s_	USCS	*w*_L_/*w*_P_ [%]	*e*_min_/*e*_max_	Organic [%]
Field soil	0.65	8.5	2.62	SW-SM	37.3/35.5	N/M *	2.7
Standard sand	0.60	0	2.63	SP	N/A *	0.625/0.919	-
White pebble	3.4	0	2.78	SP	N/A	N/M	-

* N/A: not applicable, N/M: not measured

## Data Availability

The data presented in this study are available on request from the corresponding author.

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
