# Peer review of "Preliminary Study on Application and Limitation of Microbially Induced Carbonate Precipitation to Improve Unpaved Road in Lateritic Region"

_materials, 2022, doi:10.3390/ma15207219_

Round 1

Reviewer 1 Report (Previous Reviewer 2)

The manuscript is interesting in the MICP to improve unpaved road. However, there are some issues that need to be considered.

1.  in Fig. 2, the evidence of optimal ratio (25% field soil +75% standard sand) is insufficient. This is closely related to the dry density and permeability of samples.

2. in Fig.6,on a scale of 340kPa to 403kPa, the MICP improvement in strength is very limited.

3. in Fig.8,SEM showed that there was little difference between b and f.

Author Response

Dear reviewer

Authors have prepared the response to reviewers' comments in a single file. Please take a look the file.

Reviewer 2 Report (New Reviewer)

The authors have presented a laboratory study on MICP treatment for the purpose of preserving unpaved road surface. MICP is an emerging soil improvement technology, still at the research level, and proposing MICP for unpaved road is relatively a new application of MICP. The work is significant and worth to be published. However, the following comments must be carefully addressed before acceptance.

1. The authors have chosen a field soil, which seems to be well-graded soil. Previous researchers reported that the applicability of MICP is limited to fine soils and well-graded soils, due to issues related to permeability and microbial transport. I couldn't see any discussion on this matter.

2. Of course, when the sand/ coarse materials are mixed, the treatment becomes possible...! I am concerning about the cost of sand in real applications, because authors proposed this work for the developing country. Also, mechanical mixing requires energy and equipment. Is the proposed strategy still economical? A discussion on cost is needed.

3. The authors witnessed a lower strength while treating by 2 M solution compared to 1 M solutions. What are the reasons for this?

4. Authors should correct the following throughout the manuscript; while writing OD600, the 600 should be in subscript. For example, see in Line 459.

5. One major concern of this application, the production of ammonium by products, has not been discussed anywhere. Manuscript should slightly talk about the negative side of the proposal, together with the possible mitigation methods. This makes the standard of the manuscript high. For example, ammonium by products can be eliminated/ mitigated via extraction of lasting solutions, through struvite precipitation, use of zeolite, etc. Please refer related MICP works performed in recent past and enhance the discussion in this regard.

6. What was the rate of spraying used herein? any influence in spraying rate? 

Author Response

Dear reviewer

Authors have prepared the response to reviewers' comments in a single file. Please find the file.

Reviewer 3 Report (New Reviewer)

Overall, I think it is a good work with interesting results and I believe it could be a good contribution to the scientific community. The experimental design is adequate and the explanation of the results as well. Although, if possible, I suggest combining the results and discussion sections. The conclusions are clear.

However, I believe there are some aspects that could be improved and I detail them below:

- Abstract: Regardless of whether or not the results of the experiments are as expected, I recommend always focusing on a positive message and how your results contribute to the scientific community. In fact your results are interesting.

- Introduction: The message is not clear. I recommend reformulating and focusing on aspects better related to the experiments performed. For example, address the importance of studying different types of soils or something about operating parameters.

L48-52: The information provided therein is important and has no bibliographic reference.

Mixing Method: This concept generates confusion, please be more precise.

- Materials and Methods: This section is very extensive, it should be more concise. Only the analytical methods used and the experimental methodology should be discussed. Some results are shown and even discussions are made in sections. If you worked on soil characterization, I recommend including it as results. 

Author Response

Dear reviewer

Authors have prepared the response to reviewers' comments in a single file. Please find the file.

Reviewer 4 Report (New Reviewer)

In this paper, the use of microorganisms was studied on an unpaved road system consisting of field soil. A series of soil tests were conducted using different chemical concentrations to measure the precipitation efficiency, intensity and quality of the building materials and to see the feasibility of the proposed method. Readers in this field will be interested in the content. All comments have been reasonably explained and changed. The text is well structured and the conclusions are reliable. This article can be accepted.

Author Response

Dear reviewer

Authors have prepared the response to reviewers' comments in a single file. Please find the file.

Reviewer 5 Report (New Reviewer)

It is a good paper. Some points were listed and are required for improvements.

1. Some words were written in a different font, and please correct the font.

2. It is necessary to provide a flowchart of the method with phases/steps and also describe such phases/steps. The flowchart is essential to readers' understanding clearly how the research was performed.

3. Insert a paragraph showing the contribution of the main findings from your research in this subject's current state of the art.

4. The language must be revised to native English to correct grammatical errors.

5. It was challenging to understand and follow some data in your paper. For example, you often use X = number (along the text). Would it be possible to try to include tables?

5. Please, provide improvements in the figure legends (axis x and y).

6. Include more MPDI references, for instance:  https://doi.org/10.3390/su12156281

7. The conclusion should contain the following items:

Contribution to the body of knowledge,

Contribution of the study to the practice,

Limitations of the research,

Policy implications,

Further studies recommendations.

Author Response

Dear reviewer

Authors have prepared the response to reviewers' comments in a single file. Please find the file.

Round 2

Reviewer 1 Report (Previous Reviewer 2)

  • This article is acceptable

Reviewer 2 Report (New Reviewer)

Authors addressed all the comments. The manuscript can be accepted.

This manuscript is a resubmission of an earlier submission. The following is a list of the peer review reports and author responses from that submission.

Round 1

Reviewer 1 Report

Microbially induced carbonate precipitation (MICP) is an emerging soil improvement technology. A series of soil testing was performed with varying chemical concentrations to measure precipitation efficiency, strength, and quality for construction material and see the feasibility of the proposed methods on the unpaved road. Which provided some interesting results. Id like to give some comments on this paper.

1. The relationship between water content and dry density under different proportions should be supplemented.

2. The device diagram is not clear in Fig. 2.

3. This paper shows that the solidification effect of surface spraying method is mainly concentrated on the surface due to blockage. So it is suggested to supplement the vertical section of corresponding soil sample.

4. The relationship between the penetration resistance of the micro penetrometer and the unconfined compressive strength should be supplemented.

Reviewer 2 Report

This article is about the study of MICP to improve unpaved road, including three method for implementation. However, the article is not outstanding in terms of methodology and experimental techniques. The conclusions of this study have been presented in previous publications.